# Genetic influence on within-person longitudinal change in anthropometric traits in the UK Biobank

Kathryn E. Kemper [1] ✉, Julia Sidorenko[1], Huanwei Wang[1], Ben J. Hayes [2], Naomi R. Wray[1,3], Loic Yengo[1], Matthew C. Keller [4], Michael Goddard[5,6] & Peter M. Visscher [1,7] ✉

The causes of temporal fluctuations in adult traits are poorly understood. Here, we investigate the genetic determinants of within-person trait variability of 8 repeatedly measured anthropometric traits in 50,117 individuals from the UK Biobank. We found that within-person (non-directional) variability had a SNP-based heritability of 2–5% for height, sitting height, body mass index (BMI) and weight ($P \leq 2.4 \times 10^{-3}$). We also analysed longitudinal trait change and show a loss of both average height and weight beyond about 70 years of age. A variant tracking the Alzheimer's risk $APOE\text{-}\mathcal{E}4$ allele (rs429358) was significantly associated with weight loss ($\beta = -0.047$ kg per yr, s.e. 0.007, $P = 2.2 \times 10^{-11}$), and using 2-sample Mendelian Randomisation we detected a relationship consistent with causality between decreased lumbar spine bone mineral density and height loss ($b_{xy} = 0.011$, s.e. 0.003, $P = 3.5 \times 10^{-4}$). Finally, population-level variance quantitative trait loci (vQTL) were consistent with within-person variability for several traits, indicating an overlap between trait variability assessed at the population or individual level. Our findings help elucidate the genetic influence on trait-change within an individual and highlight disease risks associated with these changes.

Genome-wide association studies (GWAS) have successfully identified 1000's of loci across the genome associated with traits and diseases (e.g. Zhou et al.[1]). Classical regression models applied in GWAS assume the variance of residuals (i.e., everything that is not explained by the SNP tested for association) are independent of genotype. However, many studies have shown associations between genetic variants and trait variance at the population level[2–6] (so-called variance Quantitative Trait Loci or vQTL), suggesting that this assumption is sometimes violated. While the biological mechanisms underlying vQTLs are poorly understood, these loci are thought to indicate the presence of unmodelled interactions[7] such as genotype x environment (GxE) or

genotype × genotype (GxG) epistatic interactions. Wang et al.[3], for example, reported several vQTL and find that these vQTL are enriched among SNPs showing genotype × age interactions (i.e. an age-dependent effect on traits).

The vQTL analysis approach typically uses cross-sectional data (which are more readily available than longitudinal data) and tests for associations between genetic variants and trait variance (e.g., by regressing SNPs on the squared centred trait value). That is, trait variance is studied at the population level and thus requires only one observation per person. An alternative is to study trait variability within an individual, e.g., the absolute deviation of repeated measurements

---

[1]Institute for Molecular Bioscience, University of Queensland, Brisbane, QLD, Australia. [2]Queensland Alliance for Agriculture and Food Innovation, University of Queensland, Brisbane, QLD, Australia. [3]Department of Psychiatry, University of Oxford, Oxford, UK. [4]Institute for Behavioral Genetics, University of Colorado, Boulder, CO, USA. [5]Faculty of Veterinary and Agricultural Science, University of Melbourne, Parkville, VIC, Australia. [6]Biosciences Research Division, Agriculture Victoria, Bundoora, VIC, Australia. [7]Big Data Institute, Li Ka Shing Centre for Health Information and Discovery, Nuffield Department of Population Health, University of Oxford, Oxford, UK. ✉e-mail: k.kemper@uq.edu.au; peter.visscher@uq.edu.au

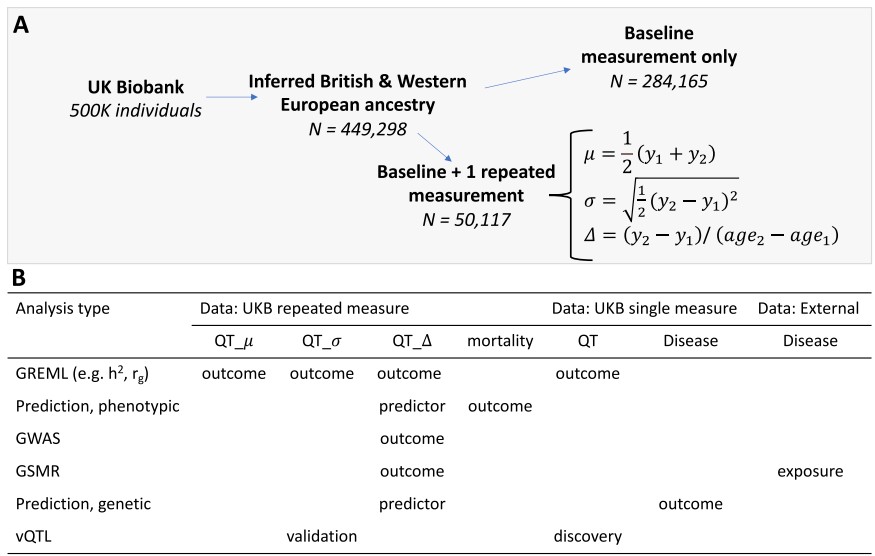

**Fig. 1 | Summary of the experimental design and techniques used in this manuscript. A** The UK Biobank (UKB) sample was split into 2 sets of unrelated individuals with genetically inferred European ancestry, those with follow-up measurements or those with a baseline measurement only. Individuals with follow-up measurements were restricted to 2 measurements and we calculated three derived phenotypes per person, their trait mean ($\mu$), trait deviation ($\sigma$) and rate of trait change ($\Delta$); **B** Summary of UK Biobank and external data, and analyses. Types of analyses include estimation of genetic parameters using the GREML approach (i.e., estimation of SNP-based heritability and genetic correlations), phenotypic and genetic prediction of disease, a genome-wide association study (GWAS), generalised summary-based Mendelian Randomisation (GSMR) analysis and replication of variance quantitative trait loci (vQTL). For each analysis type the datasets used for the dependent (i.e. discovery, predictor or exposure) and independent (i.e. outcome or replication) variables are noted, in either the UKB single or repeated measurement sub samples. Note the three derived phenotypes ($\mu$, $\sigma$, $\Delta$) were for quantitative traits (QT).

from the same individual, if such data are available. Within-person variability is robust to confounding with unmeasured factors which are different between people but relatively constant within a person (e.g., diet or activity level). Comparing vQTL at the population-level to within-individual variability could help distinguish between different types of interaction effects for vQTL because, for example, we would not expect GxG effects to contribute to variability within an individual[7].

The simple approach of using the standard deviation of repeated measurements for individuals ignores a possible directional change over time and therefore an alternative to treating the variability within an individual as a phenotype is to study the direction of change of a phenotype over time. For example, growth trajectories are commonly used as a diagnostic tool in children[8], and height and weight loss are expected with advanced age in adults[9,10]. Some studies of growth in children[11] and cross-sectional trajectories of BMI in adults[12] suggest genetic correlations significantly less than one across ages. This is equivalent to saying that there is genetic variation in the rate of trait change with time. In this study we use both approaches to investigate repeated observations. We draw on repeated measures to assess the genetic basis of within-person variability for 8 traits in the UK Biobank. We use 2 approaches to assess within person variability—absolute deviations within an individual and variation in the rate of change with age (Fig. 1). In the first approach, we assess the absolute difference of repeated measurements to estimate the SNP-based heritability of trait fluctuations. In the second approach, we explicitly model age-related trait change in height, sitting height, weight and BMI. With these measures we estimate the SNP-based heritability of trait change and explore a 2-stage random regression approach to model genetic variation as a continuous function of age. We quantify the association between trait change and all-cause mortality, and between trait-change and disease outcomes. Finally, we test if the (absolute) within-person variability can replicate population-level vQTL to gain insight into the contribution of within-individual variability to population-level vQTL.

We find that the genetic contribution to within-person variability trait is small, but significant, and report specific associations between trait-change and health outcomes.

## Results

### Overview

We analyse 2 repeated measures of 8 anthropometric traits: weight, height, sitting height, waist and hip circumference, fat percentage, body-mass-index BMI, and waist-to-hip ratio; where a schematic of the data types and analyses are given in Fig. 1. After quality control (Methods), we included 50,117 unrelated individuals ($\pi < 0.05$, where $\pi$ is the estimated genomic relationship between pairs) of genetically inferred British & Western European ancestry aged between 40 and 79 years at their baseline measurement (sample sizes for other ancestries were too small to conduct these analyses). The second trait measurement was taken an average of 7.6 years later than the first (range of 2.1-13.6 years). The correlation between the 1st and 2nd measures was very high, exceeding 0.9 for most traits (Supplementary Table S1). To analyse the repeated measures we calculated (i) the mean of the two measurements [i.e. $\mu = \frac{1}{2}(y_1 + y_2)$; where $y_1$ and $y_2$ are the raw measurements of a trait], (ii) the (absolute) variability from the mean [i.e. $\sigma = \sqrt{\frac{1}{2}(y_2 - y_1)^2}$] and (iii) the (directional) rate of trait-change over time [$\Delta = (y_2 - y_1)/(age_2 - age_1)$, where $age_1$ and $age_2$ correspond to the age at which the two measurements were taken]. We first analysed the genetic influence on an individual's mean and absolute deviation using a SNP-based genetic correlation analysis[13]. This analysis suggested potential genetic influence on within-individual trait variation for a subset of 4 traits (height, weight, sitting height and BMI), and we took these traits forward for detailed analysis of trait trajectories over time. To analyse trajectories, we corrected each individual's trait-change ($\Delta$) for the population average and then conducted a range of genetic analyses on the mean-adjusted trait-change. Analyses included a SNP-based genetic correlation analysis between the trait-mean and rate of change, a genome-wide association study of trait-change and an analysis of associations between trait-change and disease. Finally, we used the complement set of 284 K individuals in the UK Biobank with baseline observations (only) to discover population-level vQTL and attempted to replicate these vQTL using the (absolute) variability within an individual.

### Genetic influence on (absolute) variability within an individual

We used a bivariate GREML analysis[13] to estimate the SNP-based heritability and genetic correlation between the within-person mean trait measurement ($\mu$) and absolute trait deviation ($\sigma$) for 8 anthropometric traits (Table 1). The SNP-based heritability estimate for the mean of the two trait measurements were in line with previous reports[14]. We expect the heritability of the mean of 2 measurements to be slightly increased relative to estimates from a single measurement, in the order of 1.05, due to a slight reduction in the measurement error associated with the repeated observations on a trait with highly correlated measurements (Supplementary Note 1). The SNP-based heritability estimates for the within-person absolute deviation were significantly different from zero for height (0.024 s.e. 0.007, $P = 2.0 \times 10^{-4}$), sitting height (0.020 s.e. 0.007, $P = 2.4 \times 10^{-3}$), BMI (0.054 s.e. 0.007, $P = 9.0 \times 10^{-15}$) and weight (0.053 s.e. 0.007, $P = 2.6 \times 10^{-14}$; Table 1). Estimates were also nominally significant for hip circumference ($P < 0.05$), although this did not pass Bonferroni correction for multiple testing across traits ($P < 0.05/8$). A significant and strong genetic correlation between the within-person mean and absolute-deviation was observed for BMI (0.872 s.e. 0.052, $P = 1.1 \times 10^{-63}$) and weight (0.799 s.e. 0.050, $P = 1.4 \times 10^{-56}$), consistent with a mean-variance association, but the relationship was not significantly different from zero for height or sitting height. This implies that the genetic association between an individual's mean height and variability is not strong.

### Longitudinal change in height, sitting height, weight and BMI

An alternative approach to studying non-directional deviations from the mean trait measurement is to model the trait's change with age ($\Delta$, Fig. 1). This approach relaxes the assumption, compared to the analysis of absolute deviation above, that the repeated measurements are of the same trait with a genetic correlation of one between measurements. To study longitudinal trait change we first model the average population trait-change, and then determined the genetic influence on deviations from the population average. Modelling the population average is important as our data are both cross-sectional and longitudinal. For example, 2 identical height measurements for an individual in their 40's will have a different interpretation compared to two identical measurement for someone in their 70 s because, on average, a person's height is expected to decrease with age[10].

The population average trait change with age was modelled by fitting sex-dependent linear regressions on the (mean corrected) age and age[2] for height, sitting height, weight and BMI. Results show the rate of change in height is always negative and linearly dependent on age for both males and females (i.e., individuals shrink and the rate of shrinking increases with age, Fig. 2A, Supplementary Table S2). The linear effect on rate of change is significantly greater for females as compared to males ($P = 1.4 \times 10^{-7}$; $b_f = -0.005$ cm/yr, s.e. $1.4 \times 10^{-4}$; $b_m = -0.004$ cm/yr, s.e. $1.4 \times 10^{-4}$). Thus, height decreases with age in both sexes, and the total height loss is greater in females than in males. Integration of the sex-dependent regressions with respect to age makes this effect more obvious and shows, on average, height loss from 40 to 80 years is about 2.88 cm in females; and about 2.58 cm for males (Fig. 2B, Supplementary Note 2). The population average trait change in sitting height follows a similar pattern to (standing) height but with larger effects. Thus, overall height loss from 40 to 80 was predicted to be greater for sitting height, compared to standing height, and greater in females compared to males (i.e., 3.61 cm for females, 3.28 cm for males; Supplementary Fig. S1, Supplementary Note 2).

The population average change in weight showed a significant sex-dependent linear relationship with age ($P = 2.3 \times 10^{-8}$; $b_f = -0.016$ kg/yr, s.e. $6.8 \times 10^{-4}$; $b_m = -0.010$ kg/yr, s.e. $7.0 \times 10^{-4}$) and, for females only, a significant quadratic effect of age[2] ($a_f = 5.8 \times 10^{-4}$ (kg/yr)[2] s.e. $7.6 \times 10^{-5}$, Supplementary Table S2). Thus, there was a steady (males) or slight increase in weight (females) until the age of about 55 followed weight loss thereafter (Fig. 2C). Integration of the sex-dependent regressions with respect to age and evaluation at the relevant ages predicted a relative weight loss, on average, of 2.74 kg for females and 2.88 kg for males between the ages of 55 and 80 years (Fig. 2D). There was a similar trend in the population mean change for BMI although, because of concurrent reductions in height, the point of maximum BMI was slightly delayed to around 60 years (Supplementary Fig. S1). Overall, the population average change in BMI with age was relatively small (<1 kg/cm[2] over the 40-year period).

### Genetic control over longitudinal trait change

Each individual's trait-mean or rate of trait-change with age was corrected for the sex-dependent population average age-related mean or trait-change (see Methods). These new sex and age-corrected traits were used to estimate the SNP-based heritability and genetic correlations (between the mean and rate of change) using a bivariate GREML analysis in GCTA[13]. The SNP-based heritability for the trait-mean were very similar to those previously presented and the SNP-based heritability of the sex and age-corrected rate of change was about half that reported for the within-person absolute deviation of the measurements (Table 1). The SNP-based heritability on the rate of change for height (0.015 s.e. 0.006), weight (0.035 s.e. 0.007) and BMI (0.031 s.e. 0.007) were significantly greater than zero ($P < 0.05$). There was a significant genetic correlation between mean BMI and the rate of BMI change (0.163 s.e. 0.059; $P = 0.006$) but not, notably, between mean height and rate of height change. Similar to the relationship between the trait mean and absolute deviation, this suggests that there is not a strong genetic relationship between mean adult height and rate of height change.

We also wanted to estimate the genetic correlation between different ages, where a genetic correlation less than one suggests different genetic factors influence the trait. We started by using the bivariate GREML results, that is the 2 × 2 variance-covariance matrix for the trait mean and rate of trait-change (i.e. slope), in a 2-stage random regression. The random regression approach allows for trait variation to be modelled as a continuous function of age. Then, for example, the genetic correlation between any two ages can be determined using the parameter estimates from the 2 × 2 variance-covariance matrix. Note our analysis is '2-stage' as our input into the bivariate GREML analysis are mean-corrected phenotypes. For example, we found the genetic variance of weight at (mean-corrected) age $x$ is given by:

$$g(x) = 46.1 + 0.172x + 0.020x^2$$

where 46.1 is the genetic variance estimate for the mean, 0.020 is the genetic variance estimate for the slope and 0.172 is twice the genetic covariance between the mean and the slope. The residual variance and genetic covariance between any two ages can be calculated in a similar way (see Supplementary Note 3 for details). We estimate genetic correlations less than one between young (50 years) and older (70 years) ages for BMI and weight (e.g., for weight $r_g = 0.918$ s.e. $0.017$ $P = 1.7 \times 10^{-6}$; Supplementary Note 3). However, we also observed a strong concave shape for the error variance and this may indicate potential problems with the model. This could arise from the extrapolation of observations from a relatively short period (average follow-up 7.4 years) to a span of over 40 years, concurrent with imposing a linear change on the trait. This observation does bring these results into question, but we detail the approach to highlight its applicability to longitudinal data. Future applications might address the current limitations by expanding the interval between measurements and/or increasing the number of records per person.

An alternative to the 2-stage random regression, and similar to[12], is to use the age-corrected observations and cross-sectional data to estimate genetic correlations between ages in a series of bivariate analyses. We divided the 284 K UK Biobank participants of inferred

**Table 1 | Bivariate GREML estimates of variance components between the (within-person) mean and two measures of within-person variability**

| Trait | $h^2_{SNP}$ | | | | | $r_g$ | s.e. | P |
|---|---|---|---|---|---|---|---|---|
| | mean | s.e. | variability | s.e. | P-value | | | |
| A. Within-person trait mean and (absolute) trait deviation | | | | | | | | |
| height | 0.527 | 0.008 | 0.024 | 0.007 | $2.0 \times 10^{-4}$ | 0.094 | 0.053 | 0.081 |
| BMI | 0.256 | 0.008 | 0.054 | 0.007 | $8.8 \times 10^{-15}$ | 0.872 | 0.052 | $1.1 \times 10^{-63}$ |
| weight | 0.284 | 0.008 | 0.053 | 0.007 | $2.6 \times 10^{-14}$ | 0.799 | 0.050 | $1.4 \times 10^{-56}$ |
| body fat % | 0.242 | 0.008 | 0.000 | 0.007 | 0.966 | — | — | — |
| waist cir. | 0.226 | 0.008 | 0.011 | 0.006 | 0.103 | — | — | — |
| hip cir. | 0.246 | 0.008 | 0.009 | 0.006 | 0.157 | — | — | — |
| waist:hip | 0.200 | 0.008 | 0.000 | 0.006 | 0.969 | — | — | — |
| sitting height | 0.456 | 0.008 | 0.020 | 0.007 | $2.4 \times 10^{-3}$ | 0.078 | 0.063 | 0.213 |
| B. Within-person trait mean and rate of trait change | | | | | | | | |
| height | 0.524 | 0.008 | 0.015 | 0.006 | 0.023 | −0.130 | 0.072 | 0.072 |
| BMI | 0.254 | 0.008 | 0.035 | 0.007 | $3.4 \times 10^{-7}$ | 0.163 | 0.059 | 0.006 |
| weight | 0.284 | 0.008 | 0.031 | 0.007 | $3.0 \times 10^{-6}$ | 0.089 | 0.059 | 0.129 |
| sitting height | 0.452 | 0.008 | 0.010 | 0.006 | 0.125 | — | — | — |

Within-person variability was assessed with either A. the non-direction (absolute) trait deviation or B. the age- and sex- adjusted rate of trait change for 2 repeated measures of up to 8 anthropometric traits. Variance components are the SNP-based heritability ($h^2_{SNP}$) and genetic correlation ($r_g$) between the mean and the measure of variability, where genetic correlations are reported only when $h^2_{SNP}$ of both traits significantly greater than zero ($P < 0.05$). Estimates are shown with standard errors (s.e.), and chi-squared tests (with 1 df) were used to calculated unadjusted P values.

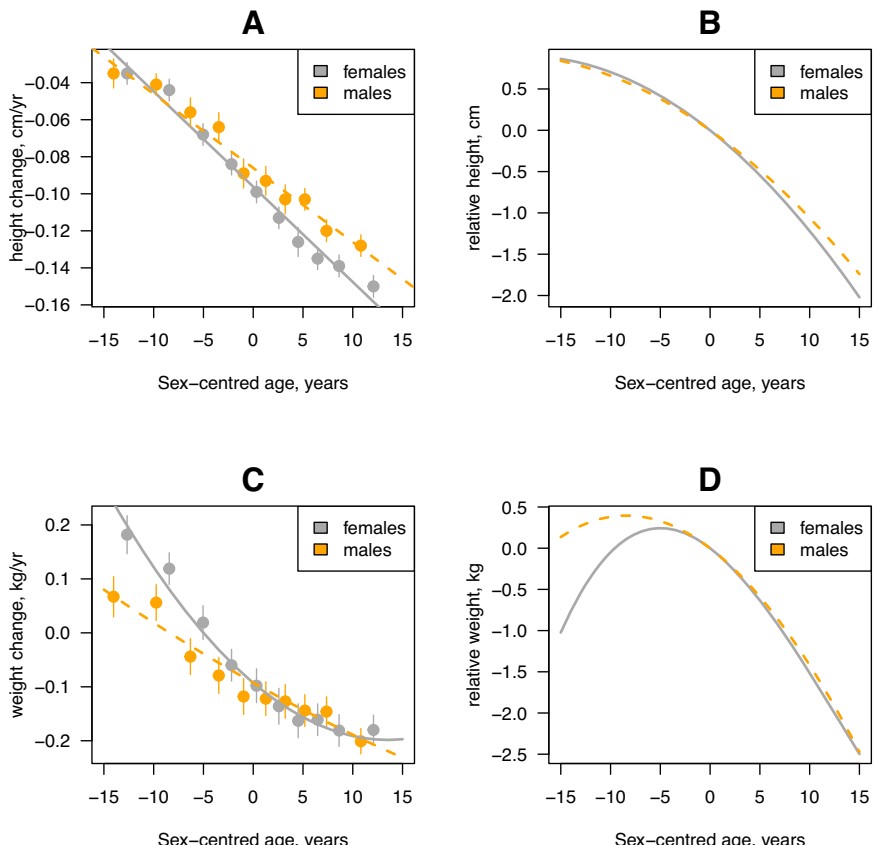

**Fig. 2 | Within-person trait change for height and weight is dependent on sex and age.** Panels show rate of height **A** and weight **C** change, where lines indicate the sex- and age-dependent polynomial fitted to the data. Points and vertical bars (95% CI) indicate the mean value of the rate of trait-change in 10 (approximately) equal age groups for the average age of measurement for females ($N = 25,759$) and males

($N = 24,313$). Also shown are the curves for cumulative height (**B**, cm) and weight (**D**, kg) change obtained by integrating, with respect to age, the sex-specific regressions shown in **A**, **C**. The average age of measurement is 59.0 and 60.3 years for females and males, respectively.

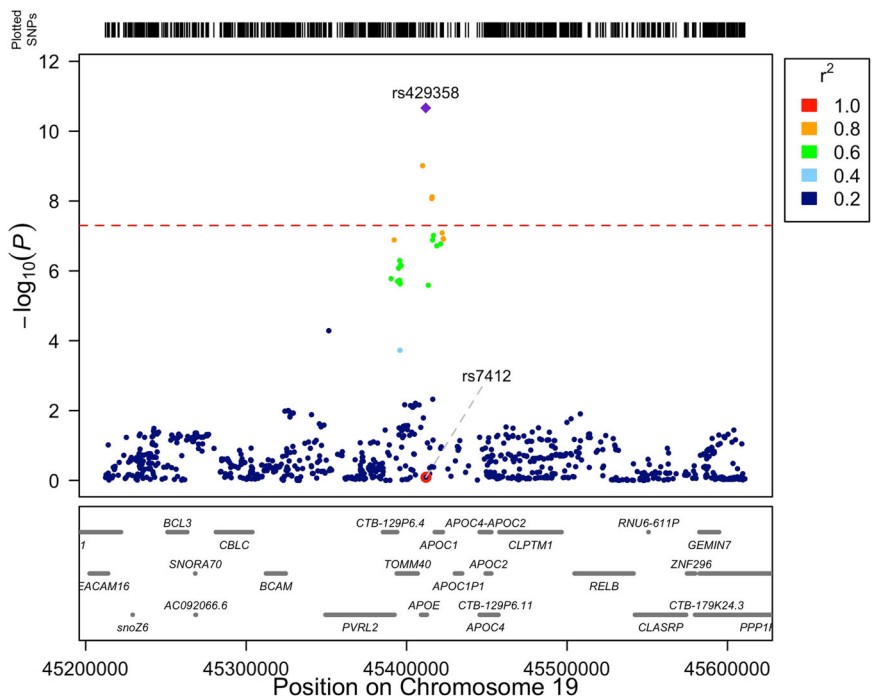

**Fig. 3 | The association between weight change per year and SNP near the *APOE* gene on chromosome 19.** The two highlighted SNP (rs429358 and rs7412) are in the *APOE* gene and define the *APOE* alleles[2] $\mathcal{E}2$, $\mathcal{E}3$, $\mathcal{E}4$ and $\mathcal{E}3r$, where the *APOE*-$\mathcal{E}4$ allele is the largest known risk locus for Alzheimer's disease[19,20]. The rs429358 SNP tracks *APOE*-$\mathcal{E}4$ almost perfectly as *APOE*-$\mathcal{E}3r$ is rarely observed (Supplementary Table S5). Shown are the chi-squared (with 1 df) test statistics for the linear association between sex and age-corrected weight change per year ($N = 49,999$), and the linkage disequilibrium (LD, $r^2$) between rs429358 and all other variants 2 Mb region. LD was calculated using 284,165 unrelated UK Biobank participants with European ancestry.

European ancestry with a single observation into 5 age subgroupings (i.e., 40–49 years, 50–54 years, 55–59 years, 60–64 years and 65–69 years) and estimated the genetic correlation between age groups using Haseman-Elston regression (Supplementary Table S3). These analyses indicated genetic correlations significantly less than one between the youngest and oldest age groups for BMI and weight. For example, the genetic correlation between 40–49 year old individuals and those 65–69 years was 0.919 (s.e. 0.023) for weight and 0.926 (s.e. 0.023) for BMI, which was significantly different from 1.0 ($P = 3 \times 10^{-4}$ and $P = 0.001$, respectively).

**Phenotypic association of trait-change with all-cause mortality**

There is evidence of a U-shaped distribution for the association between weight change and all-cause mortality[15,16]. That is, both weight loss and weight gain in later life are (phenotypically) associated with increased risk of death. We hypothesised that change in height, sitting height, weight and BMI are associated with all-cause mortality, and tested for linear and quadratic associations in our data. We found strong phenotypic associations between trait change and all-cause mortality for all 4 traits (Supplementary Table S4). The strongest linear effect was with change in height ($b = -0.217$, s.e. 0.028, $P = 1.9 \times 10^{-14}$), where height loss was associated with increased mortality. Weight and BMI showed strong quadratic or U-shaped distributions, where increased mortality was associated with either weight gain or loss (e.g., for weight, $a = 0.044$, s.e. 0.006, $P = 2.8 \times 10^{-13}$). There was also a smaller quadratic effect for height-change, where increased mortality was associated with height gain (i.e., relative to the mean, so maintenance of height) or loss ($a = 0.037$, s.e. 0.007, $P = 3.7 \times 10^{-7}$).

**A genome-wide association study for rate of trait-change**

To further investigate the genetic influences on health outcomes, we next aimed to identify individual variants associated with the rate of trait-change. We tested up to 6,493,789 imputed sequence variants (MAF > 0.01; missingness <0.01) from the UK Biobank for associations with the age-corrected rate of change for height, sitting height, weight, and BMI in the 50,117 individuals with 2 repeat observations. There were 4 variants that reached genome wide significance ($P < 1 \times 10^{-8}$) for rate of change for weight and BMI.

The 4 variants identified for rate of change for weight and BMI included the top variant rs429358 (Fig. 3; $P = 2.16 \times 10^{-11}$ for weight, $P = 7.7 \times 10^{-11}$ for BMI). The C allele of rs429358 with the C allele of rs7412 defines the *APOE*-$\mathcal{E}4$ allele, where *APOE*-$\mathcal{E}4$ is associated with increased risk of late onset Alzheimer's disease[17,18]. In this study, the C allele at rs429358 is associated with loss of weight ($\beta = -0.047$ kg per yr, s.e. 0.007) and BMI ($\beta = -0.016$ kg/m$^2$ per yr, s.e. 0.003). The rs429358 variant explains 0.09% of the variance in the rate of weight change. Interestingly, the second variant of the *APOE*-$\mathcal{E}4$ allele (rs7412) was not associated with any rate of change traits ($P > 0.05$, Fig. 3). When we investigated the two variants defining the *APOE* alleles we found the C allele of rs429358 almost perfectly tracks the *APOE*-$\mathcal{E}4$ allele as the 4th potential allele at *APOE* ($\mathcal{E}3r$) is rarely observed[19] (Supplementary Table S5).

**Causal relationships with disease**

We investigated the putative causal relationships between the rate of change traits (outcome) and risk to disease (exposure) using the Generalised Summary-data-based Mendelian Randomisation (GSMR) method[20]. The GSMR method uses the effect of multiple SNP identified from an exposure trait GWAS ($b_{zx}$) and their effect on an outcome trait ($b_{zy}$) to estimate the causal effect of the exposure on the outcome ($b_{xy} = b_{zy}/b_{zx}$). The approach requires several genetic variables (i.e., > 10 independent SNP) with strong genome-wide significant associations with the exposure to maximise power. Thus we used large meta-analysed GWAS as exposures in the analyses and selected GWAS representing risk to coronary artery disease (CAD)[21], osteoporosis[22] and Alzheimer's disease[23].

We found lumbar spine and femoral neck bone mineral density (BMD), two clinical predictors of osteoporosis, to have GSMR results

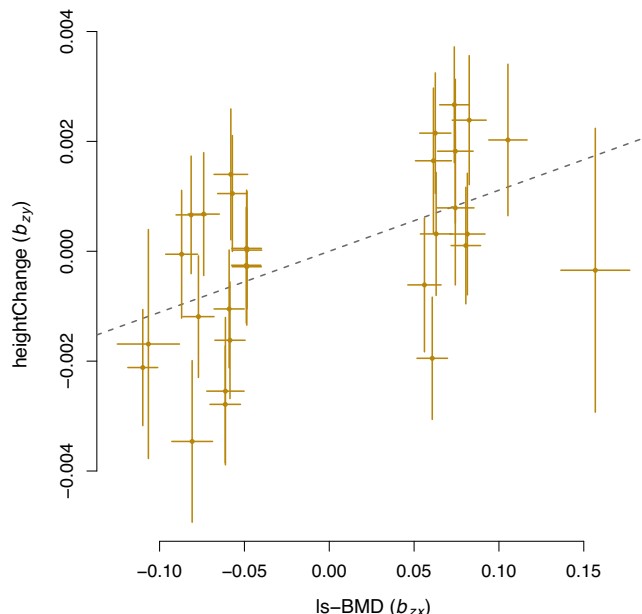

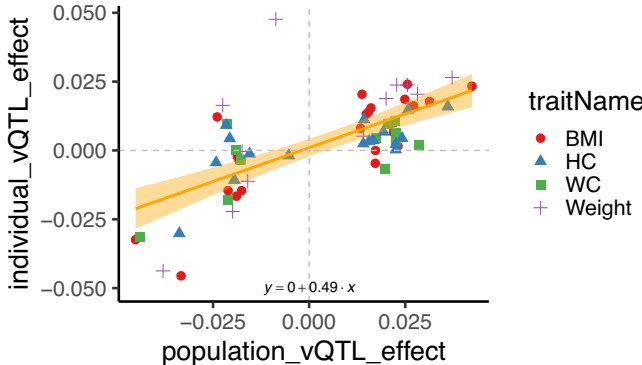

**Fig. 5 | Variance-controlling quantitative trait loci (vQTL) effects estimated either at the population level or using within-person variability.** Loci were ascertained for significant population-level effects ($P < 1\times10^{-8}$) for body mass index (BMI, red circle), hip circumference (HC, blue triangle), waist circumference (WC, green square) and weight (purple cross), and their effects re-estimated using within-individual variability. The estimated regression across all traits is shown with 95% CI ($\beta = 0.49$, s.e. = 0.07), and each trait is shown independently in Supplementary Fig. S3.

**Fig. 4 | Two-sampled Mendelian Randomisation suggests a causal relationship between risk of low lumbar spine bone mineral density (ls-BMD) and height loss.** The effect of lumbar spine bone mineral density (ls-BMD) on long-itudinal height-change using 30 ls-BMD associated SNPs from Estrada et al.[24] as instrument variables. The regression line from GSMR[22] ($b_{xy} = 0.011$, s.e. 0.003, $\chi_1^2 = 12.75$, $P = 3.5 \times 10^{-4}$) indicates that increased osteoporosis risk (i.e. decreased ls-BMD) is causative for greater height loss with age. Each point represents the average effect of an individual SNP on either ls-BMD (x-axis, $N = 31,800$) or rate of height-change (y-axis, $N = 50,072$) with bars indicating the standard error of the estimates.

consistent with a causal association with rate of height change ($P < 0.005$). The strongest relationship suggested that decreased lumbar spine BMD was causally associated with greater height loss ($b_{xy} = 0.011$, s.e. 0.003, $P = 3.5 \times 10^{-4}$, based on 30 BMD-associated SNPs; Fig. 4). We also estimated a causal effect of Alzheimer's risk on weight change ($b_{xy} = -0.034$, s.e. 0.005, $P = 2.3 \times 10^{-10}$, based on 64 AD-associated SNPs), but this result was dependent on the many SNP located on chromosome 19 (i.e., near the *APOE* locus). Since horizontal (biological) pleiotropy can bias Mendelian randomization estimates[20], and the APOE locus was highly significant for weight-change, we conducted a sensitivity analysis excluding chromosome 19 for Alzheimer's risk and weight change. After excluding chromosome 19 we found a similar magnitude of effect but, with fewer SNPs included in the model, the causal effect of Alzheimer's risk on weight change was not sig-nificantly different from zero ($b_{xy} = -0.022$, s.e. 0.014, $P = 0.12$, based on 23 genetic instruments). There was no evidence for a significant causal effect of CAD risk on any rate-change trait ($P > 0.05$).

### Genetic predictors and their relationship with disease
An alternative, complementary, approach to using GSMR is to associ-ate polygenic scores (PGS) of the rate-change traits with the disease of interest. This approach is dependent on disease prevalence in the UK Biobank, rather than large external disease GWAS in the 2-sampled MR analysis above where we could test for causal associations between the exposure and outcome. The PGS has the advantage of combining information from all SNP and not being reliant on a handful of strongly associated SNP. However, the PGS analysis is a simple association statistic which cannot make statements about causality.

We created a PGS for each rate-change trait and aimed to test the association between the PGS and disease diagnosis in an independent subset of the UK Biobank ($N = 284,165$) using logistic regression. For the disease traits, we defined disease (see Methods) based on ICD-10

codes corresponding to major osteopathic fractures, lumbar spine fractures, CAD and AD. We determined sufficient power to detect associations between the rate-change traits and disease under some realistic assumptions (Supplementary Note 4). However, the incidence of lumbar spine fractures and AD in the UK Biobank samples was low (<1%), meaning that the PGS needed to have a large effect to detect an association at the required significance threshold. For the traits with higher prevalence (all-cause mortality, major osteopathic fractures and CAD) there was > 80% power to detect PGS with odds-ratio of > 1.03.

The PGS for rate-change traits were not predictive of all-cause mortality, diagnosis of CAD, Alzheimer's disease or lumbar spine fractures (Supplementary Table S7). However, there was a significant association between diagnosis of major osteopathic fractures and the PGS for rate-change in height (−0.026, s.e. 0.011, $P = 0.01$). This equates to an odds-ratio of 0.97, meaning that a 1-SD unit increase in PGS for rate-change in height (i.e., maintenance of height with age) is asso-ciated with a 3% reduction in the odds of osteopathic fractures.

### Replication of vQTL using within-individual variability
Finally, we investigated if vQTL detected using cross-sectional data at the population-level could be replicated using within-individual (absolute) variability. Population-level vQTL are typically ascribed to unmeasured interactions with the genotype, and our analysis aims to clarify potential sources of these interactions. We detected 70 inde-pendent population-level vQTL reaching genome-wide significance ($P < 1 \times 10^{-8}$) across the 8 traits using a published approach on the set of 284 K individuals with inferred British and Western European ancestry, and only baseline measurements[3] (Supplementary Data 1). These vQTL represented 54 SNP, with several SNPs identified for more than one trait (Supplementary Fig. S2). For example, rs62106258 on chromo-some 2 was associated with heterogeneous variance of four traits; weight, BMI, hip and waist circumference. There were more than 5 SNPs (per trait) identified for BMI, weight, waist and hip circumference and these were taken through to the replication stage using within-person variability.

We attempted to replicate vQTL for BMI, weight, waist and hip circumference by regressing the individual-level vQTL effect on the population-level effect and determining if the regression slope was significantly different from zero (Fig. 5). This across-vQTL replication analysis was performed because of relatively limited power to replicate individual loci. The regression slope was significantly greater than zero

($\beta = 0.49$, s.e. $= 0.07$, $P = 7.5 \times 10^{-8}$) using all SNP across traits, and each regression was also significant when traits were considered independently ($P < 0.05$, Supplementary Fig. S3). These results indicate replication of population-level vQTL using variability within an individual.

## Discussion

Studies in human genetics typically focus on genetic variation between individuals. Here, we investigate the genetic influence on variability within individuals for anthropometric traits in the UK Biobank. For some traits, we explicitly model and describe a component of this variability, i.e., the age-related directional trait change. Longitudinal data on biobank scale with associated genetic data is a data type that is likely to increase over the next decade in terms of sample sizes and traits represented. Our approach provides a framework for analysis as these data sets emerge.

We observed genetic variation in both the absolute deviation of trait measurements, and in the rate of trait change (per year) for height, weight and BMI ($P < 0.05$; Table 1). Notably, we estimate the mean height for an individual is genetically uncorrelated with within-person variability or trait-change in height ($P > 0.05$; Table 1). We find that the Alzheimer's disease risk locus (rs429358) is strongly associated with weight-loss (Fig. 3), and present evidence consistent with a causal association between increased osteoporosis risk (lumbar spine BMD) and height loss (Fig. 4). Finally, we replicate vQTL discovered at the population-level using within-person variability. This finding indicates that within-person variability can (at least in part) drive heterogeneous variance at the population-level discovered using cross-sectional data.

This study is one of the largest using genomic information to investigate age-related change in height and weight in an older cohort, and explicitly model longitudinal trait change. From 40 to 80 years, we estimate a mean reduction in height for females of about 2.88 cm and a 2.58 cm reduction in males. This is similar to a large Austrian cohort[24] and an English study which predicted a cumulative 2–4 cm decline from maximum height with age[25]. Our observations on weight are consistent with studies citing a decrease in body weight after the age of 60 years[26] which is referred to as sarcopenia (i.e. age-related loss in muscle mass and strength)[9,27]. Our results indicate the population mean loss in body weight associated with aging is <5 kg in both males and females. Since both linear and quadratic effects of age are evident on the population-mean for BMI and weight, we recommend careful consideration of the impact of population age-related effects when analysing BMI and weight using cross-sectional data.

Our study used two different sources of information, that is longitudinal and cross-sectional data, to confirm that the genetic correlation between weight measured at younger ($\leq 50$ years) and older ages ($\geq 65$ years) is significantly less than one (2-stage random regression: $r_g = 0.918$ s.e. 0.017; cross-sectional: $r_g = 0.919$ s.e. 0.023). These results are supported by Robinson et al.[12], who estimate a significant interaction between genotype and age for BMI (i.e. a GxE, where 'E' is age; they estimate $r_g = 0.56$ s.e. 0.19 between 18 and 40 year olds and individuals > 66 years). This is despite the SNP-based heritability for weight being relatively consistent across age brackets ($h^2_{SNP}$ ~ 0.27, Supplementary Table S3). The presence of genotype by age interactions may explain some of the variability between twin and family-based heritability estimates for BMI where, depending on the experimental design, GxE effects could be partitioned into the common environmental or residual component[12,28]. We observed a curvature in the environmental variance (and therefore heritability estimates) from the 2-stage random regression and propose this was caused by the extrapolation of short-term within-person estimates of trait change to a span of over 40 years. Future applications of random regression should aim for observations over a longer follow-up period to obtain more robust results.

We found trait-change over time to be associated with health outcomes. At the phenotypic level, there was a strong U-shaped association with all-cause mortality for weight and BMI change with age (e.g., for weight, $a = 0.044$, s.e. 0.006, $P = 2.8 \times 10^{-13}$; Supplementary Table S4). That is, individuals closely following the population mean age-related change had the lowest risk of death. Our results add to the debate for the role of longitudinal change in BMI with all-cause mortality, where both loss and gain of weight have been associated with increased mortality in older adults[15]. We also showed a linear and quadratic relationship between height and sitting height-change and all-cause mortality (e.g., for height, $b = -0.217$, s.e. 0.031, $P < 2 \times 10^{-16}$; $a = 0.037$, s.e. 0.007, $P < 3.7 \times 10^{-7}$). Although linear effects of height-loss and mortality are evident in previous studies[29], a quadratic effect is rarely tested and is more difficult to explain. We showed using GSMR evidence consistent with osteoporosis risk (i.e. ls-BMD) being causally associated with greater height loss with age (Fig. 4).

We identify that the C allele of the rs429358 SNP as significantly associated with the rate of weight loss over time (Fig. 3; $-0.047$ kg per yr, s.e. 0.007, $P = 2.2 \times 10^{-11}$). Venkatesh et al.[30] also identify this variant using a different approach studying longitudinal change in BMI in the UK Biobank, and it is also genome-wide significant in Jiang et al.[31] for the categorical response to 'weight change compared to 1 year ago' ($-0.018$, s.e. 0.002, $P = 9.5 \times 10^{-21}$). The rs429358 SNP is a cystine-to-arginine missense variant in the APOE gene and, with rs7412, forms the haplotype defining the APOE-$\mathcal{E}$4 allele. This allele is a well-known AD risk variant[17] and it accounts for about 20% of the SNP-based heritability in AD[32]. Few of the UK Biobank participants in our study had an AD diagnosis prior to their final measurement and so, given the risk of AD for the rs429358 carriers, our findings provide strong evidence supporting previous reports of weight loss proceeding an AD diagnosis[33]. Our initial GSMR results suggested a causal link between AD risk and weight-loss, but this result was dependent on the APOE locus which is has a large effect on both traits. Thus, we cannot make strong conclusions about early-stage AD causing weight loss (i.e., due to reduced appetite), weight loss being a predisposing factor to AD or potentially a third factor influencing both traits.

The APOE gene encodes for a glycoprotein (apoE)[34] where different isoforms exhibit different clearance rates of the amyloid-$\beta$ (A$\beta$) peptide in the brain[35] and build-up of A$\beta$ deposits is a defining characteristic of AD[36]. The APOE-$\mathcal{E}$4 allele has been associated with a range of traits, including increased risk of hypercholesterolaemia and coronary artery disease and reduced risk of obesity and type 2 diabetes[37]. However, our study finds no evidence to support an association between rs7412 with weight change ($P > 0.05$) and the rs429358 SNP tracks the APOE-$\mathcal{E}$4 allele almostly perfectly (Supplementary Table S5). Bennet et al.[38] use A$\beta$ concentration in spinal fluid and serum lipid levels to provide compelling evidence for independent action of the two variants. That is, they find that rs429358 alone mediates A$\beta$ concentration in the central nervous system and rs7412, with a minor (independent) influence of rs429358, is associated with peripheral lipid levels. We checked the results of Jiang et al.[31] to confirm that the rs429358 is strongly associated with UK Biobank participants reporting their mother's AD diagnosis and with taking cholesterol lowering medication, and that rs7412 is also associated with these traits (Supplementary Fig. S4). However, we note that the observed test statistics and linkage disequilibrium patterns are consistent with multiple causal variants for cholesterol lowering medication use in this region. Future studies examining APOE should consider these variants independently to aim for a clearer resolution of pleiotropy at this locus.

Finally, we show evidence for the replication of population-level vQTL using within-individual variation (Fig. 5, Supplementary Fig. S3). Population-level vQTL[3–6] have been consistently reported for a number of traits, most numerously for BMI[39,40]. The typical interpretation of a population-level vQTL is that the increased variance for a genotype class arises from an (unobserved) interaction effect that mixes two

normal distributions with different means. Wang et al.[3], for example, found that their population-level vQTL were enriched for genotype-by-environment (GxE) interactions. Our analyses extend these findings to explicitly include within-person variability as potential source of variability in population-level vQTL discoveries. Studying variability within individuals restricts the range of potential interacting factors as, for example, epistatic effects are fixed within an individual and cannot contribute to variability over time. The relative importance of population-level GxE effects and within-individual effect is unknown but our results, combined with the observation that age is frequently a significant GxE effect[3], suggests that closer examination of longitudinal age-related trait changes are warranted.

The UK Biobank is known to have a 'healthy volunteer' bias, and this is exaggerated by using a subset of the UK Biobank with repeated observations. For example, the all-cause mortality is estimated at 45–55% lower in the UK Biobank compared to the general population residing in the United Kingdom[41]. We observe that individuals with repeated observations have 20–50% reduction in the incidence of disease diagnosis compared individuals in the UK Biobank with only baseline observations, and there is approximately a 70% reduction in the all-cause mortality rate among these individuals (compared to UK Biobank participants with only baseline measurements, Supplementary Table S6). We conducted a GWAS based on having baseline only or follow-up measures and found loci differentiated between the two groups to be also associated with traits such as educational attainment, current smoking, sleep duration, BMI, triglycerides and walking pace (Supplementary Note 5). Sometimes these associations had effects consistent with a healthy volunteer bias (e.g. the C allele of rs2410678 was more frequent in individuals with repeated measures and decreased current smoking) but sometimes there was evidence for antagonistic pleiotropy (e.g. the G allele of rs784256 was more frequent in repeated measure individuals with the effect of increased walking pace and educational attainment, but the G allele also decreased mood and increased incidence of corneal dystrophy). There is evidence of an age-related bias in the repeated measures sample as the frequency of the AD risk allele (rs429358, Supplementary Fig. S5) decreases in individuals older than about 60 years at baseline who subsequently returned for a follow up visit. Evidence of participation bias in the UK Biobank is in line with many other reports[42]. We expect attenuation of main effects due to a healthy participation bias but caution is warranted due unpredictable the nature of collider bias[43] and pleiotropy (as described above).

We note that scale-dependent effects might influence our key results (Supplementary Note 6) and conclude that, although there potentially exists a scale which will homogenise the variances[44], we conducted our analysis on the biologically relevant scale.

In summary, we report evidence for genetic control of within-person variability for height, sitting height, weight and BMI. We show for weight and BMI evidence for a genotype-by-age interaction, where the genetic correlation between younger (≤50 years) and older ages (≥ 65 years) is less than unity. We provide evidence for weight loss associated with the rs429358 allele prior to a formal AD diagnosis, and a causative effect of osteoporosis risk on height loss. Conclusions from our study are limited by the relatively short follow-up time (average 7.6 years), and participation biases inherent in volunteer-based cohorts such as the UK Biobank. Our approach provides an analytical framework for researchers to build on as larger genetic data sets with repeated observations become available.

## Methods
### Ethical compliance
The UK Biobank study was proved by the North West Centre for Research Ethics Committee (11/NW/0382). Participants volunteered for the study and provided signed electronic consent. Details on the ethics and governance framework of the UK Biobank is available on the UK Biobank website (https://www.ukbiobank.ac.uk/media/0xsbmfmw/egf.pdf). This research is approved under the University of Queensland human ethics committee (approval number 201100173).

### UK Biobank data
The data used for this study were from the full release of the UK Biobank. Briefly, the UK Biobank is a study of about 500,000 people recruited from across the United Kingdom. Individuals have a range of physical measurements, biological samples and questionnaire-style assessments at up to four timepoints, namely at baseline (2006-2012), a first repeat assessment (2012-2013), and a first (2014 +) and repeat (2019 +) imaging visit. Genotyping data consists of 807,411 or 825,927 markers from the UKBiLEVE and UK Biobank Axiom arrays, with quality control and whole genome imputation using the Haplotype Reference Consortium panel performed by the UK Biobank team[45]. We identified approximately 450 K individuals for further analysis who passed basic quality control and ancestry filters[14]. These individuals have (i) inferred British and Western European ancestry[14], (ii) a consistent self-reported and genetic sex, (iii) had not rescinded consent, (iv) imputed genotypes, (v) born between 1937 and 1970 and (vi) aged between 40 and 69 at baseline.

### Phenotypes
Six height and weight-related traits (height, sitting height, weight, fat percentage as measured by impedance, waist and hip circumference) were extracted at the four above mentioned timepoints in the UK Biobank (Supplementary Table S1). We calculated body mass index (BMI, height/weight$^2$) and waist to hip ratio (WHR, waist/hip) at each timepoint. There were 53,373 individuals with more than one assessment and from this set we removed (i) one member in each pair of relatives (where relatives were defined by their genomic relationship ($\pi$), and $\pi > 0.05$), and (ii) individuals failing additional quality control measures (see below). Genomic relationships were determined with 1.1 M imputed HapMap3 SNP (minor allele frequency, MAF > 0.01) using GCTA (v1.93.2 beta)[31]. Quality control measures removed individuals with height, weight, waist or hip circumference observations > 5 SD from their individual mean, where the SD was the population average standard deviation of observations per individual for each trait. Individuals without a recorded date of measurement, a baseline assessment or with <2 observations after quality control were also removed. The final set consisted of 50,117 individuals.

Most individuals (82%) in our set of individuals had only 2 observations. Hence, we created a final dataset with exactly 2 observations for each person. In this set, we used the baseline assessment plus either the 2$^{nd}$ imaging or 1$^{st}$ recall assessment (if 2$^{nd}$ imaging visit was not available); or both observations if only 2 were available. The mean time to follow-up in the dataset was 7.6 years with standard deviation of 2.6 years. For each individual we then calculated 3 new phenotypes for each trait, the within person (i) mean [$\mu = \frac{1}{2}(y_1 + y_2)$; where $y_1$ and $y_2$ are the 1$^{st}$ and 2$^{nd}$ measurements of a trait], (ii) the absolute deviation [$\sigma = \sqrt{\frac{1}{2}(y_2 - y_1)^2}$] and (iii) rate of trait change [$\Delta = (y_2 - y_1)/(age_2 - age_1)$, where $age_1$ and $age_2$ correspond to the age of two measurements].

### Additive genetic variance associated with the mean and within-person variability
To examine the genome-wide influence of additive genetic variance on variability, we conducted a SNP-based heritability analysis using the 50,117 individuals with exactly 2 repeated measures. We fitted a model to the within-person mean and absolute deviation as follows:

$$Z = mu + sex + batch + age + centre.1 + centre.2 + ageDiff + e \quad (1)$$

where $Z$ is the individuals mean ($\mu$) or absolute deviation ($\sigma$), mu is the overall mean, sex is the genetic sex (male or female), batch is the

genotyping batch (106 levels), age is the age at baseline measurement (as a factor, 30 levels), centre.1 (22 levels) and centre.2 (5 levels) are the assessment centres of the $1^{st}$ and $2^{nd}$ measurements; ageDiff is the age difference in days between measurements and $e$ is the residual. Residuals were standardised N(0,1) within sex. A bivariate GREML analysis using GCTA (v1.93.2 beta)[31] was used to estimate the SNP-based heritability and genetic correlation between the trait-mean and absolute deviation. In GCTA we fitted 25 principal components from[14] as covariates. The genomic relationship matrix is described above (i.e., that used to identify unrelated individuals) and was built with 1.1 M HapMap3 SNP with MAF > 0.01.

## Replication of vQTL using within-person variability
We conducted a discovery genome-wide vQTL study using an independent subset of the UK Biobank from the individuals with single measurements ($N = 284,165$). These individuals were unrelated ($\pi < 0.05$) to each other and were also unrelated ($\pi < 0.05$) to the 50,117 individuals with repeat measurements. We extracted the 6 height and weight-related phenotypes (height, sitting height, weight, fat percentage, hip and waist circumference), calculated BMI and WHR, and fitted the following model to the data:

$$y = mu + sex + yob + batch + age + centre + e \quad (2)$$

where $y$ was the measured phenotype, mu, sex, batch and age are as defined in model [1]; yob is year-of-birth as a factor (34 levels), centre is the assessment centre where the measurement was taken (22 levels) and $e$ is the residual. We normalised residuals within sex and fitted 25 principal components from[14] as covariates to test 8,538,964 autosomal imputed markers (MAF > 0.01, biallelic, Hardy-Weinberg equilibrium > $1 \times 10^{-5}$, info score > 0.3, missing genotype rate > 0.05) for associations with heterogeneous error variance. We followed the approach of Wang et al.[3] and as implemented in the OSCA (v0.46) software[46]. Genotypes were hard-called prior to analysis using PLINK2 (--hard-call 0.1)[47]. Independent genome-wide significant ($P < 1 \times 10^{-8}$) markers were identified using the '--clump-r2 0.01' and '--clump-kb 5000' options in PLINK (v1.90p)[47]. Similar to Wang et al.[3], we detected 8,786 loci that were genome-wide significant across 6 traits (BMI, weight, fat percentage, waist and hip circumference and waist:hip ratio; $P < 1 \times 10^{-8}$), a total of 70 independent trait-loci genome-wide significant markers (Supplementary Table S3), or 54 independent loci across the genome. A breakdown of the number of loci per trait, and the number of loci associated with multiple traits is shown in Supplementary Fig. S2. Traits where there were > 5 genome-wide significant loci identified (BMI, WC, HC, weight) were taken forward to a replication phase using within-individual variability.

The replication of effects was tested using a regression approach. That is, the estimated population-level vQTL effects were used as independent variables (x-axis) to predict or explain the within-person variability (y-axis). The expectation for replication is a regression slope ~ 1. Population-level vQTL effects were estimated for the genome-wide significant loci using the $z^2$ approach of Yang et al.[4], where $z$ is the z-score and $z^2$ assesses variance. Phenotypes were squared residuals from model [2], standardised to a unit normal within sex. GWAS was conducted using the fastGWA-lr option in GCTA (v1.93.2 beta)[31] with 25 principal components from[14] fitted as covariates. Estimation of within-person variability effects followed a similar procedure using GCTA and fitting principal components from[14] as covariates, where phenotypes were the squared standard-deviation ($\sigma_i^2$) residuals from model [1], standardised to a unit normal.

## Longitudinal analysis
The longitudinal analysis used the within individual trait-mean, the rate of trait change and the average age of measurement of height, weight, sitting height and BMI. The age of measurement was first corrected for

the mean within each sex, then we fitted a model as follows:

$$Z = mu + sex + age + sex.age + age^2 + sex.age^2 \quad (3)$$

where $Z$ is the individual's mean ($\mu$) or rate of change ($\Delta$), mu is the overall mean, sex is the effect of being male or female, age and $age^2$ are the linear and quadratic effects of age, and sex.age and $sex.age^2$ are the interactions between these effects and sex. Terms without significant effects on the outcome ($P > 0.01$) were dropped from the model before the creation of age and sex-corrected residuals for the trait mean and rate of change. We took the integral of final equations with respect to age to infer the relative trait change for each trait (Supplementary Note 2). A bivariate GREML analysis using GCTA (v1.93.2 beta)[31] was conducted (as above) to estimate the SNP-based heritability and genetic correlation between the age- and sex-corrected trait mean and rate of change for each trait. In GCTA we fitted 25 genotypic principal components from[14] as covariates.

Full details of the random regression results and transformations applied can be found in Supplementary Note 3. The supporting bivariate analysis divided the 284,165 individuals with single records into 5 age groups based on the age at measurement, treated each age group as a trait and estimated the genetic correlation between the age groups with the Haseman-Elston (--HEreg) option in GCTA (v1.93.2 beta)[31]. The genomic relationship used was constructed with 1.1 M HapMap3 SNP (MAF > 0.01)[14] and we also fitted the 25 principal components[14] as covariates. Standard errors were estimated using the jackknife approach in GCTA. Traits were residuals from (2) for height, weight, BMI and sitting height.

## Association of trait-change with disease & mortality
The associations used logistic regression and either the age- and sex-corrected rate of change as the phenotypic predictor, or a PGS derived from the GREML analysis as the genetic predictor. Predictors were standardised to a unit normal prior to the regressions. To create the PGS, we used the --reml-pred-rand and --snp-blup options in GCTA (v1.93.2 beta)[31] to estimate SNP effects, and then predicted the PGS for the independent subsample of the UK Biobank, i.e. individuals with only a single measurements ($N = 284,165$).

All-cause mortality was for any recorded death in the registry (UKB field 40023). Osteoporotic outcomes[48] were defined by an ICD-10 code (UKB field 41202) for lumbar spine fracture (S320) or major osteoporotic fracture affecting the hip, vertebrae, humerus or wrist (M80, S120-S122, S127, S220-S221, S320, S422-S424, S525-S526, S620-S621, or S72). CAD was defined following Jiang et al.[49] using ICD-10 codes for myocardial infarction (I21-I23, I24.1 or I25.2) and coronary revascularisation (K401-K404, K411-K414, K451-K455, K491, K492, K498, K499, K502, K751-K754, K758-K759). We diagnosed Alzheimer's disease cases as of August 2021 (assessed via RAP) based on the following criteria: individuals with (i) available date of first occurrence of G30 (Alzheimer's disease) or F00 (dementia in Alzheimer's disease, UKB fields 131036 and 130836); (ii) ICD-10 diagnosis (UKB field 41270) or G30 or F00 or (iii) self-reported (UKB field 200002, instance 0-3) dementia/Alzheimer's/cognitive impairment.

## Genome-wide association study of rate of trait-change
We tested up to 6,493,789 imputed sequence variants (MAF > 0.01, missingness <0.01) from the UK Biobank for an association with the age and sex-corrected rate of trait change for height, sitting height, BMI and weight. Associations used the --fastGWA-lr option in GCTA (v1.93.2 beta) and we fitted 25 principal components[14] as covariates.

## GSMR analysis
We downloaded publicly available summary statistics for CAD[50], femoral neck and lumbar spine BMD[22] and Alzheimer's disease[23] (excluding UK Biobank and 23andMe samples), and conducted a

GSMR[20] analysis using GCTA (v1.93.2beta)[31]. To maximise power, we tested for exposure to disease risk as causative for trait-change. Briefly, GSMR identifies independent variants for the disease GWAS reaching genome-wide significance ($P < 5 \times 10^{-8}$, $r^2 < 0.01$) using a reference set of genotypes, where we used the UK Biobank sample with repeated measures ($N = 50,117$) as the reference. The corresponding variants from the trait-change GWAS are extracted and the SNP effects for the outcome ($b_{zy}$) regressed on the exposure ($b_{zx}$) to estimate the causal effect of the exposure on the outcome ($b_{xy} = b_{zy}/b_{zx}$). We did not conduct GSMR analyses using the trait-change traits as the exposure trait because there were few independent genome-wide significant SNP to select as instrumental variables.

### Reporting summary

Further information on research design is available in the Nature Portfolio Reporting Summary linked to this article.

## Data availability

Data from this study is available from the UK Biobank. Data access policies (http://www.ukbiobank.ac.uk/register-apply/) and a description of the genetic data (http://www.ukbiobank.ac.uk/scientists-3/genetic-data/) are available from the UK Biobank website. We downloaded the genome-wide summary statics for Alzheimer's disease (excluding UK Biobank and 23andMe samples) reported in Wightman et al.[23] from the Complex Traits Genetics lab website (https://ctg.cncr.nl/documents/p1651/PGCALZ2ExcludingUKBand23andME_METALInverseVariance_MetaAnalysis.txt.gz), the male and female pooled summary statistics reported in Estrada et al.[22] for femoral neck and lumbar spine bone mineral density from the genetic factors for osteoporosis consortium website (http://www.gefos.org/sites/default/files/GEFOS2_FNBMD_POOLED_GC.txt.gz and http://www.gefos.org/sites/default/files/GEFOS2_LSBMD_POOLED_GC.txt.gz), and the CADIoGRAM meta-analysis statics reported in Schunkert et al.[50] from the CARDIoGRAMplusC4D consortium website (http://www.cardiogramplusc4d.org/media/cardiogramplusc4d-consortium/data-downloads/cardiogram_gwas_results.zip). Data and scripts to reproduce figures and tables are provided in the Source data provided with this paper. Summary Statistics for BMI, weight, height, and sitting height rate-change; plus the case-control analysis of single and repeat measures participants in the UK Biobank are available for download on the website (https://cnsgenomics.com/content/data). Source data are provided with this paper.

## Code availability

Software programs used in this study are all publicly available; PLINK2 and PLINK v1.9 can be downloaded from Christopher Chang's website (https://www.cog-genomics.org/plink/), GCTA and OSCA from the Yang lab website (https://yanglab.westlake.edu.cn/software/gcta/#Overview), R from the CRAN website (https://cran.r-project.org/) and Rstudio from the posit website (https://posit.co/products/opensource/rstudio/).

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

## Acknowledgements

This research has been conducted using the UK Biobank Resource under project 12505. This research was supported by the Australian National Health and Medical Research Council (1113400; N.R.W. and P.M.V.), the Australian Research Council (FL180100072; P.M.V.), the National Institutes of Health (MH100141 and MH130448; M.C.K.) and the National Health and Medical Research Council (Investigator Grant 1173790; N.R.W.).

## Author contributions

K.E.K. and P.V.M. conceived and designed the analyses. K.E.K. performed the analyses. J.S., H.W., B.J.H., N.R.W., L.Y., M.C.K. and M.E.G. contributed to data quality control and advised on the analysis. K.E.K. and P.V.M. wrote the manuscript, with the participation of all authors.

## Competing interests

The authors declare no competing interests.
