## [Peer Review File · Nature Communications]

Genetic influence on within-person longitudinal change in anthropometric traits in the UK BiobankREVIEWER COMMENTS

Reviewer #2 (Remarks to the Author):

The manuscript by Kemper et al. investigates the genetic determinants of changes in human traits over time. Overall, the manuscript is well written and I do not have any major concerns regarding data analyses. Unfortunately, the study does not identify very many genetic variants and very strong effects and the main results that are highlighted in the study are more related to the effects of weight gain/loss and height loss. However, the same topic has previously been investigated by different approaches, and my main concern with this manuscript is therefore the lack of novel insights arising from the present study. There are, however, a few additional analyses that could be done, as highlighted below.

Specific comments:

1. The authors use the measurements from multiple assessments in the UK Biobank. Most individuals (82%) had only 2 observations, which limits the ability to make any conclusion about how the traits change over time. However, there was a recent study (pre-print): <https://pubmed.ncbi.nlm.nih.gov/36711652/>, where primary-care data from UK Biobank were used to study the longitudinal change in BMI in the UK Biobank (which is also cited in the current manuscript). Why was not that data also included in this study? That would most likely increase the number of participants as well as the number of measures per participant. Longitudinal data are often analyzed using splines and/or by creating cluster profiles of longitudinal data. The few observations per individual would probably be a limitation for such an approach, so getting access to more observations per individual would dramatically increase the impact of the work.
2. The underlying causes of change in trait values could be more carefully investigated. For example, osteoporosis causes weakened bones and can cause height loss. This is more pronounced in females after menopause when the estrogen levels drop and is the likely explanation for the larger drop in height in women compared to men. This could be discussed, but also investigated using, for example, Mendelian randomization (MR) for mediation analyses. Instead, the authors use a PGS for the rate of change and test if it is associated with musculoskeletal conditions which is not an appropriate method for addressing causality and neither for drawing any conclusion with regards to the underlying biology. However, with the few variants identified in the GWAS, an MR approach might lack power.
3. The other major finding that is highlighted is that weight loss is associated with the rs429358 allele (the APOE Alzheimer's disease [AD] risk allele). This result is not novel (as the authors acknowledge) and they state that they "cannot distinguish between cause-and-effect". This is, of course true based on the analyses done, but it would be possible to do some more analyses to sort out the causal relation, e.g. As suggested in my previous comment.
4. There is a discussion on the age-related bias on the repeated measures in UK Biobank and the authors also show that such bias cause the individuals included in this study to be healthier with regards to musculoskeletal health, endocrine diagnoses, all caused mortality and with lower rates of the AD risk allele. I am, however, lacking a more comprehensive discussion/estimation of how this bias can influence the results of this study. How does this bias change with age?

Reviewer #3 (Remarks to the Author):

Dear colleagues,

Kemper et al, presented here a comprehensive approach to study the longitudinal change of several anthropometric traits, analyzing the within-person variability in the UKBB, leading to (1) the quantification of within-person variability over time and its contribution to the observed population-level residual variation, and (2) the identification age-related genetic influence of trait change over time associated to health outcomes. The authors also presented new interesting modelling approaches

to analyze age-dependent traits.

The presented work, focused mainly in height and weight, both highly polygenic traits and well explored by relevant works for the environmental impact, will help TO understand, TO explore, TO quantify and TO map the so called hidden heritability in a more wide using common datasets, exploring the power of the array-derived heritability measures, and to understand the effect of the biases existing in large existing genome cohorts. The work and the methods exposed by Kemper et al, will be significant to the complex diseases genomics fields developing, adding tools for interpretation of the hidden heritability in complex traits.

I suggest some major/minor points to be considered for reviewing.

Major points

The paper is well written in general but lacks of clarity in its narrative and redaction, and I believe that this should be ameliorate. An additional effort in editing (e.g. some repeated words, line 198, line 477) and re-writing some parts (as the 2-stage random regression description in lines 188 and ahead) should be ameliorated for a better description of the work and the reached conclusions, allowing reader to get conclusions by himself/herselve.

The authors are quite clear about the study limitations, with the methods and tools used for the analysis (e.g. low heritability, cohort bias). However, I will like the authors incorporate these limitations in other parts of the paper, more visible; being more specific and clear during the presentation of the summary work to avoid confusion about the significance and impact of this variability observed in these measured traits that, as the authors shown, is low. Highly expectancies from initial abstract is not well supported by results and conclusions. For instance, the mention about PGS in the abstract offers some kind of disconnection with the goal and results of the study.

I think that the interpretations of the GWAS for rate of trait change, still being very appealing, with a plausible connection, and statistical sound data, are over interpreted for the impact on healthy aging populations, even if limitation about hyper healthy bias in the UKBB used dataset in pointed in the last part of the conclusion. Is no clear to me, reading the text, the possible effect of the healthy bias and the APOE results (e.g significant association supporting the involvement of loss of weight and AD). As reader, I would like to know, how the calculated heritability of the trait change (low) is related to this results? And last, when reading I will prefer always to know, while reading the main text, the estimated impact of associations (OR and CI instead that the P value) to have a better idea of the results.

The methodology is sound, but perhaps overwhelming for most common interesting readers in the field. An Image summary abstract will help to visualize the presented workflow, sometimes confusing. In some parts there is an excessively detail in the man text (2 stage random regression, from line 188), that should be ameliorated to improve reading and reaching of the presented conclusions by the authors. In concrete, this paragraph in line 188, about random regression seems that this section is very extended and refers to a Supplementary note 3, that is empty.

The supplementary information is extended. Perhaps a reduction an improved clarity should be provided in the main text, with detailed information on the supplementary material, as commented before.

There are some minor points.

Please pay attention and review some paragraphs that are not well written, and are difficult to understand, or seems that lacks some words, or have repeated words in the text. This will help the reading to understand better.

In lines 41-52, when describing vQTLs, I will appreciate a better precision in the use of the term regarding the bibliography used. Seems from the bibliography a few confusing.

In lines 54-63, when describing that within-person variability is consistent within a person, making reference specifically to the diet, I think that is no very true since diet is always changing during life of a person (in your study the assessment period of time was 7 years approx.) is and we cannot assume there is few variations in this point.

Some of the tables, as the trait change (TavbleS3), I think, that merit be in the main text, since are part of the conclusions of the work, and not in the supplementary data.

Results and Figure 3 from phenotypic and PGS, and comparisons stated in the text are not clear. In

figure 3, why ratio are expressed relative to the 3rd quintile. All- cause mortality and M diagnoses are mixed with Phenotypic traits and PGS, maybe four figures will, be clearer here to compare results. Is confusing and is not explained.

We thank the reviewers for their helpful comments on our manuscript. We address each of their comments below. Key changes to the manuscript in response to the reviewers' comments are highlighted (in yellow) in the manuscript. In addition, we have made smaller changes throughout the manuscript to improve clarity and correct other minor errors.

REVIEWER 2:

The manuscript by Kemper et al. investigates the genetic determinants of changes in human traits over time. Overall, the manuscript is well written and I do not have any major concerns regarding data analyses. Unfortunately, the study does not identify very many genetic variants and very strong effects and the main results that are highlighted in the study are more related to the effects of weight gain/loss and height loss. However, the same topic has previously been investigated by different approaches, and my main concern with this manuscript is therefore the lack of novel insights arising from the present study. There are, however, a few additional analyses that could be done, as highlighted below.

We thank the reviewer for their time to read and consider our manuscript and for making helpful and constructive comments. We have taken steps to address their comments by conducting a number of major additional analyses. We would like to highlight our completely novel findings, namely the estimation of a genetic correlation between the trait-mean and trait-change for height, the linking of longitudinal trait change to disease outcomes, and the replication of variance controlling loci (vQTL) discovered at the population level using variability within individuals. We also present a novel approach to estimate genetic correlations between ages using standard software (i.e. bivariate GREML).

Specific comments:

1. The authors use the measurements from multiple assessments in the UK Biobank. Most individuals (82%) had only 2 observations, which limits the ability to make any conclusion about how the traits change over time. However, there was a recent study (pre-print): <https://pubmed.ncbi.nlm.nih.gov/36711652/>, where primary-care data from UK Biobank were used to study the longitudinal change in BMI in the UK Biobank (which is also cited in the current manuscript). Why was not that data also included in this study? That would most likely increase the number of participants as well as the number of measures per participant. Longitudinal data are often analyzed using splines and/or by creating cluster profiles of longitudinal data. The few observations per individual would probably be a limitation for such an approach, so getting access to more observations per individual would dramatically increase the impact of the work.

We thank the reviewer for their comments. We read and cited the mentioned preprint paper in our original submission. Indeed, this preprint was uploaded to medRxiv as we were finalising our manuscript for submission. That preprint paper finds the same key GWAS locus, namely the APOE-E4 risk variant as associated with weight loss. We also note in the revised manuscript that the APOE-E4 locus (rs429358) was previously identified in the Jiang et al. (2019) GWAS of the UKB for the question, 'weight change compared to 1 year ago' (P = 9.5e-21, see https://yanglab.westlake.edu.cn/data/ukb_fastgwa/imp/pheno/2306). We decided to retain

our use of the assessment center repeated measured data because new extensive analysis of the primary care data would, in our opinion, deviate from our core aims and would take a completely new research direction.

We would like to re-emphasize that our paper goes well beyond a simple GWAS of the rate-change traits. We investigate different aspects of longitudinal trait change including genetic correlations between different ages, absolute or directional trait change and its relationship with the mean trait value, and the association of trait-change and disease.

2. The underlying causes of change in trait values could be more carefully investigated. For example, osteoporosis causes weakened bones and can cause height loss. This is more pronounced in females after menopause when the estrogen levels drop and is the likely explanation for the larger drop in height in women compared to men. This could be discussed, but also investigated using, for example, Mendelian randomization (MR) for mediation analyses. Instead, the authors use a PGS for the rate of change and test if it is associated with musculoskeletal conditions which is not an appropriate method for addressing causality and neither for drawing any conclusion with regards to the underlying biology. However, with the few variants identified in the GWAS, an MR approach might lack power.

We agree with the reviewer. Questions regarding causality between the rate-change traits and disease are important to dissect further. The power of MR analyses, assuming a 2-sample MR, is dependent on the strength of the association between the exposure and the SNP, and the most powerful approaches use several independent SNP strongly associated with the exposure trait. In our case, we have only a single independent variant ($P < 1e-8$) for BMI/weight rate-change and this would not be sufficiently powerful for a MR analysis (i.e. the variant explains only 0.09% of the phenotypic variance for weight-change).

However, we have considered the issue of causality and included a GSMR (generalized summary-based Mendelian Randomisation) analysis in the revised manuscript using large-scale external GWAS of coronary artery disease (CAD), femoral neck and lumbar spine bone mineral density (fn-BMD, ls-BMD; as diagnostics of osteoporosis) and Alzheimer's Disease. These analyses test 'reverse causality', that is if genetic risk to the disease is causal for trait-change. We believe that this approach overcomes the aforementioned power issues as the summary statistics are from large external GWAS. For example, using GSMR we find that risk for low ls-BMD is causal for longitudinal height-loss ($P = 3.5e-4$). Also we find a causal relationship between AD risk and weight-loss, but find that this relationship is dependent on chromosome 19 SNP (i.e. influenced by APOE locus).

As discussed below, we have also refined our diagnosis of disease in the PGS analysis.

3. The other major finding that is highlighted is that weight loss is associated with the rs429358 allele (the APOE Alzheimer's disease [AD] risk allele). This result is not novel (as the authors acknowledge) and they state that they "cannot distinguish between cause-and-effect". This is, of course true based on the analyses done, but it would be possible to do some more analyses to sort out the causal relation, e.g. As suggested in my previous comment.

As stated above, we now include GSMR analysis to examine the causal relationship between AD and weight-loss. However this finding seems dependent on the APOE locus. A key assumption of MR analysis is that the outcome (weight-change) is only associated with the exposure (AD risk) via the genetic instruments. Unfortunately weight-change is also associated with the APOE-E4 locus, meaning that inferences about causality can be biased. These analyses are now included in the revised manuscript.

4. There is a discussion on the age-related bias on the repeated measures in UK Biobank and the authors also show that such bias cause the individuals included in this study to be healthier with regards to musculoskeletal health, endocrine diagnoses, all caused mortality and with lower rates of the AD risk allele. I am, however, lacking a more comprehensive discussion/estimation of how this bias can influence the results of this study. How does this bias change with age?

We have now included a new case-control GWAS (Supplementary Note 5) to highlight the potential effects of participation bias on our results and discuss these results further, with two clear examples. We expect attenuation of the main effects if healthier individuals are more likely to return for a follow-up visit. However, we highlight that sometimes these effects are unpredictable for pleiotropic loci. For example, we find that the G allele of rs784256 is frequent in follow-up individuals. This allele is also associated with educational attainment and walking pace (i.e. presumably with positive effects on health) but also decreased mood and incidence of eye disease (i.e. unfavorable effects). We hope that this additional GWAS and discussion addresses the reviewer's comments.

Reviewer #3 (Remarks to the Author):

Dear colleagues,

Kemper et al, presented here a comprehensive approach to study the longitudinal change of several anthropometric traits, analyzing the within-person variability in the UKBB, leading to (1) the quantification of within-person variability over time and its contribution to the observed population-level residual variation, and (2) the identification age-related genetic influence of trait change over time associated to health outcomes. The authors also presented new interesting modelling approaches to analyze age-dependent traits.

The presented work, focused mainly in height and weight, both highly polygenic traits and well explored by relevant works for the environmental impact, will help TO understand, TO explore, TO quantify and TO map the so called hidden heritability in a more wide using common datasets, exploring the power of the array-derived heritability measures, and to understand the effect of the biases existing in large existing genome cohorts. The work and the methods exposed by Kemper et al, will be significant to the complex diseases genomics fields developing, adding tools for interpretation of the hidden heritability in complex traits.

We thank the reviewer for their time considering our paper and for helpful and constructive comments.

I suggest some major/minor points to be considered for reviewing.

Major points

The paper is well written in general but lacks of clarity in its narrative and redaction, and I believe that this should be ameliorate. An additional effort in editing (e.g. some repeated words, line 198, line 477) and re-writing some parts (as the 2-stage random regression description in lines 188 and ahead) should be ameliorated for a better description of the work and the reached conclusions, allowing reader to get conclusions by himself/herselve.

We thank the reviewer for this comment. We have now re-organised the results section to add further subheadings and clarify the findings. We hope this improves the readability of the manuscript. For example, we now have headings of phenotypic associations with trait-change, causal relationships with disease for the MR analysis, and a separate section on the PGS predictors. The discussion on the random regression remains under the genetic control of longitudinal change heading but we have shortened the paragraph, and added some extra detail to improve clarity and separated the paragraph on the cross-sectional approach to estimation of genetic correlations. We hope these changes address the reviewer's concerns.

The authors are quite clear about the study limitations, with the methods and tools used for the analysis (e.g. low heritability, cohort bias). However, I will like the authors incorporate these limitations in other parts of the paper, more visible; being more specific and clear during the presentation of the summary work to avoid confusion about the significance and impact of this variability observed in these measured traits that, as the authors shown, is low. Highly expectancies from initial abstract is not well supported by results and conclusions. For instance, the mention about PGS in the abstract offers some kind of disconnection with the goal and results of the study.

We thank the reviewer for this comment. We have removed the reference to the PGS in the abstract and replaced it with the results of the GSMR analysis results. Further, and also in response to reviewer #2's comments, we have refined our disease definitions to be more specific. This means for some traits that we had limited power to detect an association with the PGS – and this is now detailed in supplementary note 4. Supplementary note 4 details our expectations for a trait-change PGS to detect an association with disease through a combination of theory and simulation. For example, we show that we have good power to detect associations with diseases with a frequency of > 3% and having an OR > 1.03.

I think that the interpretations of the GWAS for rate of trait change, still being very appealing, with a plausible connection, and statistical sound data, are over interpreted for the impact on healthy aging populations, even if limitation about hyper healthy bias in the UKBB used dataset in pointed in the last part of the conclusion. Is no clear to me, reading the text, the possible effect of the healthy bias and the APOE results (e.g significant association supporting the involvement of loss of weight and AD). As reader, I would like to know, how the calculated heritability of the trait change (low) is related to this results? And last, when reading I will prefer always to know, while reading the main text, the estimated impact of associations (OR and CI instead that the P value) to have a better idea of the results.

We hope that that the new GSMR results (as discussed above under point 2. of reviewer #2) provide a stronger link between disease and longitudinal trait change. We have expanded our discussion on the healthy volunteer bias in the UKB, and included the case-control GWAS for the number of repeated measures as a more explicit example of the effects of healthy volunteer bias (Supplementary Note 5). The most likely outcome of a healthy volunteer bias is the attenuation of effects, however this is difficult to quantify and sometimes unpredictable for pleiotropic loci.

We now include power calculations and simulations for the associations between the PGS and disease outcomes in the UKB (Supplementary Note 4). These calculations focus on the power to detect a given correlation between a PGS and a trait. The ability of the PGS for trait-change to capture disease risk is determined by the prediction accuracy of the PGS, the genetic correlation between traits, and the heritability of the traits in question.

We have also included effect sizes with standard errors throughout the manuscript, in addition to P-values.

The methodology is sound, but perhaps overwhelming for most common interesting readers in the field. An Image summary abstract will help to visualize the presented workflow, sometimes confusing. In some parts there is an excessively detail in the man text (2 stage random regression, from line 188), that should be ameliorated to improve reading and reaching of the presented conclusions by the authors. In concrete, this paragraph in line 188, about random regression seems that this section is very extended and refers to a Supplementary note 3, that is empty.

The supplementary information is extended. Perhaps a reduction an improved clarity should be provided in the main text, with detailed information on the supplementary material, as commented before.

As stated above, the middle section of the results has been re-organised to improve clarity. We have revised Figure 1 to include a more detailed summary of the data processing and analysis types used in the manuscript. The blank page previously preceding Supplementary Note 3 has been removed and we have also revised the note.

There are some minor points.

Please pay attention and review some paragraphs that are not well written, and are difficult to understand, or seems that lacks some words, or have repeated words in the text. This will help the reading to understand better.

We thank the reviewer for these comments. We have carefully reviewed the manuscript to remove repeated words and re-written or re-organised sections to improve clarity.

In lines 41-52, when describing vQTLs, I will appreciate a better precision in the use of the term regarding the bibliography used. Seems from the bibliography a few confusing.

We are not entirely sure we understand the reviewer's comments here. We have reviewed the use of the term 'vQTL' in the manuscript to be consistent and as clear as possible. vQTL simply refers to a location in the genome associated with the variance of a trait (rather than the mean). However, we appreciate, particularly in the literature, that interpretation and meaning of a vQTL is challenging as it depends on experimental design.

In lines 54-63, when describing that within-person variability is consistent within a person, making reference specifically to the diet, I think that is not very true since diet is always changing during life of a person (in your study the assessment period of time was 7 years approx.) and we cannot assume there are few variations in this point.

We thank the reviewer for this comment. We have modified this statement to say 'relatively constant within a person' because although individuals are likely to make minor changes to their diet or level of physical activity over time it seems unlikely that significant changes persist (i.e. an office worker is unlikely to start a laboring job, though they may take up running).

Some of the tables, as the trait change (Table S3), I think, that merit be in the main text, since are part of the conclusions of the work, and not in the supplementary data.

We thank the reviewer for this comment. We have now expanded Table 1 to include the data previously in Supplementary Table S3.

Results and Figure 3 from phenotypic and PGS, and comparisons stated in the text are not clear. In figure 3, why ratios are expressed relative to the 3rd quintile. All-cause mortality and M diagnoses are mixed with Phenotypic traits and PGS, maybe four figures will be clearer here to compare results. It is confusing and is not explained.

This was done to improve clarity but seems not to help. We have now removed Figure 3. Further, we only discuss phenotypic associations between trait-change and all-cause mortality; and genetic associations between the PGS and disease in the independent UKB sample.

REVIEWERS' COMMENTS

Reviewer #2 (Remarks to the Author):

I believe the authors have effectively addressed all of my concerns, and I have no additional comments to add. I commend them for their excellent work.

Reviewer #3 (Remarks to the Author):

The work is interesting but complex, so we thanks to the authors the effort to re-organise sections improving the readability of the manuscript. Furthermore, as important key in these exploratory works, we also thank the authors to have completed a clear exposition of the limitations, adding information on the power to disease's association with the PGS. Even if not a key part of the work, we also thank the new causality test using MR, adding more reasonable explanations to the PGS associations, even if recognized limitations of the approach.